# Learning Similarity Metrics for Volumetric Simulations with Multiscale CNNs

## Abstract

Simulations that produce three-dimensional data are ubiquitous in science, ranging from fluid flows to plasma physics. We propose a similarity model based on entropy, which allows for the creation of physically meaningful ground truth distances for the similarity assessment of scalar and vectorial data, produced from transport and motion-based simulations. Utilizing two data acquisition methods derived from this model, we create collections of fields from numerical PDE solvers and existing simulation data repositories, and highlight the importance of an appropriate data distribution for an effective training process. Furthermore, a multiscale CNN architecture that computes a volumetric similarity metric (*VolSiM*) is proposed. To the best of our knowledge this is the first learning method inherently designed to address the challenges arising for the similarity assessment of high-dimensional simulation data. Additionally, the tradeoff between a large batch size and an accurate correlation computation for correlation-based loss functions is investigated, and the metric's invariance with respect to rotation and scale operations is analyzed. Finally, the robustness and generalization of *VolSiM* is evaluated on a large range of test data, as well as a particularly challenging turbulence case study, that is close to potential real-world applications.

## 1 Introduction

Making comparisons is a fundamental operation that is essential for any kind of computation. This is especially true for the simulation of physical phenomena, as we are often interested in comparing simulations against other types of models or measurements from the physical system. Mathematically, such comparisons require metric functions that determine scalar distance values as a similarity assessment. A fundamental problem is that traditional comparisons are typically based on simple, element-wise metrics like the $L^1$ or $L^2$ distances, due to their computational simplicity and a lack of alternatives. Such metrics can work reliably for systems with few elements of interest, e.g. if we want to analyze the position of a moving object at different points in time, matching our intuitive understanding of distances. However, more complex physical problems often exhibit large numbers of degrees of freedom, and strong dependencies between elements in their solutions. Those coherences should be considered when comparing physical data, but element-wise operations by definition ignore such interactions between elements. With the curse of dimensionality this situation becomes significantly worse for systems that are modeled with dense grid data, as the number of interactions grows exponentially with a linearly increasing number elements. Such data representations are widely used, e.g. for medical blood flow simulations (Olufsen et al., 2000), over climate and weather predictions (Stocker et al., 2014), to the famous unsolved problem of turbulence (Holmes et al., 2012). Another downside of element-wise metrics is that each element is weighted equally, which is typically suboptimal: E.g. in the context of fluids, smoke plumes behave differently along the vertical dimension due to gravity or buoyancy, and small key features like vortices are more indicative of the fluid's general behavior than large areas of near constant flow (Pope, 2000).

In the domain of natural images, neural networks have been successfully employed for similarity models that can consider larger structures, typically via learning from the semantic meaning of the images through class labels or data that encodes our human perception. Similarly, physical systems exhibit spatial and temporal coherence due to the underlying laws of physics, from which we can derive a similarity model. To robustly learn similarity assessments of scalar and vectorial volumetric data from such a model, our work makes the following contributions:

- We propose a novel similarity model based on the entropy of physical systems. It is employed to synthesize sequences of volumetric physical fields suitable for metric learning.

- We show that our Siamese, multiscale feature network results in a stable metric that successfully generalizes to new physical phenomena. To the best of our knowledge this is the first learned metric inherently designed for the similarity assessment of volumetric fields.

- The metric is employed to analyze turbulence in a case study, and its invariance to rotation and scale are evaluated. In addition, we analyze correlation-based loss functions with respect to their tradeoff between batch size and accuracy of correlation computation.

The central application of the proposed *VolSiM* metric is the similarity assessment of new physical simulation methods, numerical or learning-based, against a known ground truth. This ground truth can take the form of measurements, higher resolution simulations, or existing models. Furthermore, the trained metric can be used as a differentiable similarity loss for physical learning problems, similar to perceptual losses for computer vision tasks.

## 2 RELATED WORK

Apart from simple $L^n$ distances, the two metrics PSNR and SSIM (Wang et al., 2004) are commonly used across disciplines for the similarity assessment of data. Similar to the underlying $L^2$ distance, PSNR shares the issues of element-wise metrics (Huynh-Thu & Ghanbari, 2008; 2012). SSIM computes a more intricate function, but it was shown to be closely related to PSNR (Horé & Ziou, 2010) and thus has similar problems (Nilsson & Akenine-Möller, 2020). Furthermore, statistical tools like the Pearson correlation coefficient PCC (Pearson, 1920) and Spearman's rank correlation coefficient SRCC (Spearman, 1904) can be employed as a simple similarity measurement. There are several learning-based metrics specialized for different domains such as rendered (Andersson et al., 2020) and natural images (Bosse et al., 2016), interior object design (Bell & Bala, 2015), audio (Avgoustinakis et al., 2020), and haptic signals (Kumari et al., 2019). Especially for images, similarity measurements have been studied in multiple settings, mainly by combining deep embeddings as perceptually more accurate metrics (Prashnani et al., 2018; Talebi & Milanfar, 2018). Similarly, such metrics can be employed for various image related tasks like super-resolution (Johnson et al., 2016) or image generation (Dosovitskiy & Brox, 2016). We additionally study the behavior of invariance and equivariance to transformations, which was targeted previously for rotational symmetries (Weiler et al., 2018; Chidester et al., 2019) and improved generalization (Wang et al., 2021).

Traditional metric learning for natural image domains typically works in one of two ways: Either, the training is directly supervised by learning from manually created labels, e.g. via two-alternative forced choice where humans pick the most similar option to a reference (Zhang et al., 2018). Or, the training is indirectly semi-supervised through images with class labels and a contrastive loss (Chopra et al., 2005; Hadsell et al., 2006). In that case, triplets of reference, same class image, and other class images are sampled, and the corresponding latent space representations are pulled together or pushed apart. We refer to Roth et al. (2020) for an overview of different image metric training strategies. Similarity metrics for simulation data have not been studied extensively yet. For smoke flow synthesis, Siamese networks for finding similar fluid descriptors have been applied (Chu & Thuerey, 2017). Um et al. (2017; 2021) used crowd-sourced user studies for the similarity assessment of liquid simulations. Simulation data was compared via a learned metric (Kohl et al., 2020) using a feature extractor with a structure similar to a classification CNN, however only considering scalar 2D data and a simple, linear distance model.

Deep learning techniques also receive attention for improving simulations, e.g. for predicting states over time (Morton et al., 2018; Kim et al., 2019), for learning via physical laws (Watters et al., 2017; Greydanus et al., 2019), and for the dynamics of complex materials (Ummenhofer et al., 2020; Sanchez-Gonzalez et al., 2020). While these approaches have potential to be combined with our metric, we focus on traditional numerical methods to acquire data in the following.

## 3 MODELING SIMILARITY OF SIMULATIONS

To formulate our methodology for learning similarity metrics that target dissipative physical systems, we turn to the fundamental quantity of entropy. The second law of thermodynamics states

that the entropy $S$ of a physical system never decreases, thus $\Delta S \geq 0$. In the following, we make the reasonable assumption that the behavior of the system is continuous and non-oscillating, and that $\Delta S > 0$. On the right, Eq. 1 is the Boltzmann equation from statistical mechanics (Boltzmann, 1866), that describes $S$ in terms of the Boltzmann constant $k_b$ and the number of microstates $W$ of a system (here an ideal gas).[1] Since entropy only depends on a single system state, it can be reformulated to take the relative change between two states into account: given a sequence of states $s_0, s_1, \ldots, s_n$, the relative entropy $\tilde{S}(s)$ is defined in Eq. 2, where $w_s$ is the monotonically increasing, relative number of microstates

$$S = k_B \log(W) \quad (1)$$

$$\tilde{S}(s) = k \log(c\, w_s) \quad (2)$$

$$D(s) = k \log(c\, w_s + 1) \quad (3)$$

$$D(s) = \frac{\log(c\, w_s + 1)}{\log(c + 1)} \quad (4)$$

defined as 0 for $s_0$ and as 1 for $s_n$. $c > 0$ is a system-dependent factor that determines how quickly the number of microstates increases. Next, we shift Eq. 2 such that the lower bound is 0 to derive a first similarity model $D$. This offset is necessary, since the properties of similarity metrics dictate that distances are always non-negative and only zero for identical states. Finally, we choose the system-dependent constant $k$ appropriately, as relative similarities do not change through linear transformations, and thus $k$ not influence the general behavior of $D$. Choosing $k = 1/(\log c + 1)$ leads to the full similarity model $D(s)$ in Eq. 4, that is bound to the range $[0, 1]$, and directly predicts the level of similarity between the reference $s_0$ and any state $s$ from a sequence.

Fig. 1 illustrates this connection between the logarithmic increase of entropy within a system and the proposed similarity model. In addition, trajectories for different $\Delta$ through this space are shown. This $\Delta$ between states corresponds to $c$ in the similarity model in Eq. 4, and is directly proportional to it: For large changes $\Delta_{inf}$ to the system, the entropy quickly increases and the states $s_1, \ldots, s_n$ contain little information, hence are quickly very dissimilar to $s_0$ (red dotted curve). For the smallest possible changes $\Delta_\epsilon$, entropy slowly increases while similarity linearly decreases, due to the continuous nature of the system (green dashed line). Ideally, a sequence of states employed for learning tasks should evenly exhibit

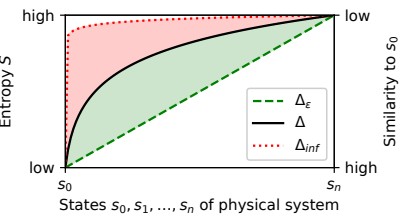

Figure 1: Idealized model of the behavior of entropy and similarity for a physical system for different changes $\Delta$.

both regimes as well as intermediate ones, as indicated by the black curve. For practicality, this should not rely on an extremely large number of states, as would be necessary for $\Delta_\epsilon$. In the following, we will refer to this property of a sequence being informative with respect to a similarity analysis as *difficulty* of a sequence with a fixed length $n$. The red area in Fig. 1 generally corresponds to sequences that are too difficult, i.e. that do not bare any similarity to $s_0$ very quickly, while the green area generally corresponds to sequences that are too easy, i.e. that barely change.

The resulting formulation for $D$ represents an idealized similarity model and does not directly work as a metric due to two reasons: 1) a metric only takes two states as an input, so estimating $c$ would be difficult, and 2) both inputs do not necessarily have to come from the same system, for example when comparing a velocity observation to a simulated ground truth. To overcome this, we formulate a semi-supervised learning problem by creating sequences $s_0, s_1, \ldots, s_n$ of different physical systems, and training a neural network $m$ to predict the distance $d$ given the reference distances $g$ as computed via $D$. This technique incorporates the underlying physical behavior, compared to adding variations in a post-process (as it is often done in the domain of images, e.g. by Ponomarenko et al. (2015)). Alg. 1 shows how $d$ and $g$ are computed: First, we estimate $c$ by evaluating Pearson's distance that indicates how quickly the states become uncorrelated, normalizing it to $[0, 1]$, and optimizing Eq. 4 for $c$ via a standard least-squares

**Algorithm 1** Computation of distance prediction $d$ of metric $m$ and ground truth distances $g$ on an ordered sequence $s_0, s_1, \ldots, s_n$.

---

**for** $i = 1$ to $n$ **do**
    $q$.append $(1 - \text{PCC}(s_0, s_i))$
$q \leftarrow$ normalize $q$ to $[0, 1]$
$c \leftarrow$ fit $\frac{log(10^c q + 1)}{log(10^c + 1)}$ for $c$

**for** $i = 0$ to $n - 1$ **do**
    **for** $j = i + 1$ to $n$ **do**
        $d$.append$(m(s_i, s_j))$
        $w_s$.append$((j - i)/n)$
$g \leftarrow \frac{log(10^c w_s + 1)}{log(10^c + 1)}$

---

[1]We do not have any a priori information about the distribution of the likelihood of each microstate in a general physical system. Thus, more generic entropy models such as Gibb's entropy or Shannon's entropy, both of which assume a non-uniform distribution of microstates, are not immediately applicable.

optimization. Then, $g$ directly follows from Eq. 4 through the fit of $c$ and the linearly increasing number of microstates $w_s$, while $d$ is computed by a pairwise evaluation of $m$.

## 4 SEQUENCE CREATION

To create sequences $s_0, s_1, \ldots, s_n$ within a controlled environment, we make use of the knowledge about the underlying physical processes: We either employ direct changes, based on spatial or temporal coherences to $s_0$, or use changes to the initial conditions of the process that lead to $s_0$. The central challenge now becomes how to find *suitable* sequences via the magnitude of $\Delta$, without knowing a priori which degree of non-linearity in the similarity model provides the most information for each setup. However, given a value of $\Delta$ and a corresponding sequence, we can compute how strongly it differs from a sequence with $\Delta_\epsilon$, as depicted in Fig. 1. This is achieved by a proxy similarity function, via computing the PCC between this proxy and a simple linearly decreasing similarity as predicted by $\Delta_\epsilon$. If the resulting correlation is high, the sequence is close to a similarity behavior as predicted by $\Delta_\epsilon$, while lower correlations indicate stronger deviations. Finally, we sample sequences based on simple correlation thresholds to acquire a suitable difficulty level. By using the mean squared error (MSE) as a proxy, we empirically determined that correlations between $0.65$ and $0.85$ work well for all cases we considered. Hence, we propose two semi-automatic iterative methods (see Fig. 2) to create data, depending on the method to introduce variations to a given state. Both methods iteratively sample a small set of sequences to calibrate $\Delta$ to a suitable magnitude and use that for the full data set. Compared to a stricter sampling of every single sequence, this method avoids potential biases in the data and is computationally more efficient as less sampling is needed.

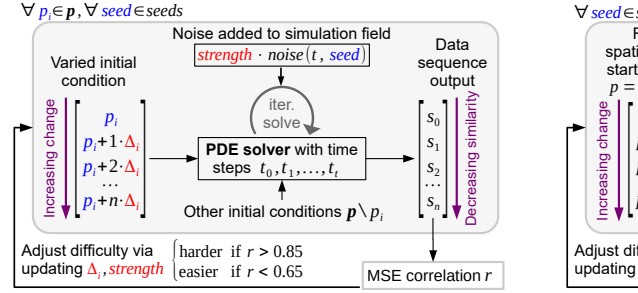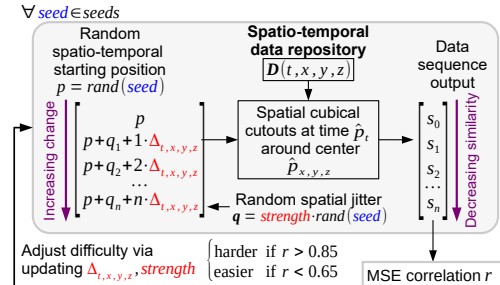

Figure 2: Iteration schemes to create data sequences of decreasing similarity. Variation from the reference state can be introduced via the initial conditions of a numerical PDE simulation (method [A], left), or via spatio-temporal data changes on data from a repository (method [B], right).

**[A] Variations based on Initial Conditions of Simulations** Given a numerical PDE solver and a set of initial conditions or parameters $p$, the solver computes a solution to the PDE over the time steps $t_0, t_1, \ldots, t_t$. To create a larger number of different sequences, we make the systems non-deterministic by adding noise to a simulation field and randomly generating the initial conditions from a given range. Adjusting *one* of the parameters $p_i$ in steps with a small perturbation $\Delta_i$, allows for the creation of a sequence $s_0, s_1, \ldots, s_n$ with decreasing similarity to the unperturbed simulation $s_0$. This is repeated for every suitable parameter in $p$, and the corresponding $\Delta$ is updated individually until the targeted MSE correlation range is reached. The global noise strength factor also influences the difficulty to some degree and can be updated.

**[B] Variations based on Spatio-temporal Coherences** For a source $D$ of volumetric spatial-temporal data without access to a solver, we rely on a larger spatial and/or temporal dimension than the one required for a sequence. We start at a random spatio-temporal position $p$ to extract a cubical spatial area $s_0$ around it. $p$ can be repeatedly translated in space and/or time by $\Delta_{t,x,y,z}$ to create a sequence $s_0, s_1, \ldots, s_n$ of decreasing similarity. Using a different position leads to new sequences, as long as the repository features enough diverse data. It is possible to add a global amount of random perturbations $q$ with a chosen strength to the positions, to further increase the difficulty.

**Data Sets** To create training data with method [A], we utilize solvers for a basic Advection-Diffusion model (`Adv`), Burger's equation (`Bur`) with an additional viscosity term, and the full Navier-Stokes equations via a Eulerian smoke simulation (`Smo`) and a hybrid Eulerian-Lagrangian liquid simulation (`Liq`). The corresponding validation sets are generated with a separate set of

random seeds. Furthermore, we use adjusted versions of the noise integration for two test sets, by adding noise to the density instead of the velocity field in the Advection-Diffusion model (`AdvD`), and overlayed noise to the background area of the liquid simulation (`LiqN`).

We create seven test sets via method [B]. Four come from the Johns Hopkins Turbulence Database JHTDB (Perlman et al., 2007) that contains a large amount of direct numerical simulation (DNS) data, where each is based on a subset of the JHTDB and features different characteristics: isotropic turbulence (`Iso`), a channel flow (`Cha`), magneto-hydrodynamic turbulence (`Mhd`), and a transitional boundary layer (`Tra`). Since turbulence contains structures of interest across all length scales, we additionally randomly stride or interpolate the query points for scale factors in $[0.25, 4]$ to create sequences of different physical size. One additional test set (`SF`) via temporal translations is based on ScalarFlow (Eckert et al., 2019), consisting of 3D reconstructions of real smoke plumes. Furthermore, method [B] is slightly modified for two synthetic test sets: Instead of using a data

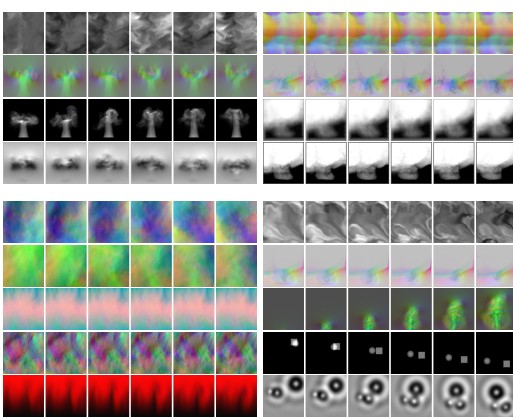

Figure 3: Example sequences from training sets (top, column-wise: `Adv`, $3\times$`Smo`, `Bur`, $3\times$`Liq`) and test sets (bottom, column-wise: $2\times$`Iso`, `Cha`, `Mhd`, `Tra`, `AdvD`, `LiqN`, `SF`, `Sha`, `Wav`).[3]

repository, we procedurally synthesize spatial fields. For that, we employ linearly moving randomized shapes (`Sha`), and randomized damped waves (`Wav`) of the general form $f(x) = cos(x) * e^{-x}$. All data was gathered in sequences with $n = 10$ at resolution $128^3$, and downsampled to $64^3$ for computational efficiency during training and evaluations. Fig. 3 shows parts of multiple example sequences via a mean projection to 2D, and App. B and C contain further details on each data set.

**Data Distribution** There are many independent factors that influence the difficulty of the created data sequences, and measuring the similarity in the generation process with a proxy function introduces additional uncertainties. In addition, our iteration schemes only calibrate $\Delta$ to a suitable magnitude, instead of enforcing stricter bounds on every single sequence. As a result, the computed PCC values from the MSE (see Fig. 2) on the full data sets exhibit a more natural, smooth distribution with controllable difficulty, instead of introducing an artificial distribution with hard cutoffs. Due to the central limit theorem, we expect a normal distribution of the PCC values that indicate the sequence difficulty. However, correlations have a fixed upper bound of 1, meaning a truncated normal distribution is a more accurate prediction. Intuitively, this corresponds to trajectories with different curvature in the similarity model in Fig. 1. In fact, we empirically determined that only training data distributions which reasonably closely follow this prediction, result in a successful training of our model. We assume, that significantly different distributions indicate unwanted biases in the data, which prevented effective learning in our experiments. Fig. 4 shows the correlation histograms of our training and validation data sets, all of which roughly follow this desired truncated normal distribution.

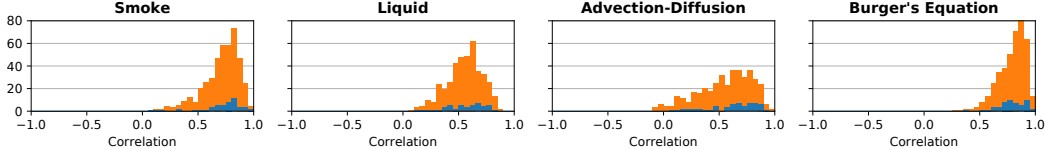

Figure 4: MSE correlation histograms of training data (orange) and validation data (blue) with a bin size of 0.05. Each histogram roughly follows a truncated normal distribution.

## 5 LEARNING A DISTANCE FUNCTION

For our method, we generally follow the established Siamese structure, that was originally proposed for 2D domains (Zhang et al., 2018; Kohl et al., 2020): First, two inputs are embedded in a latent

---

[3]Here the predominant x-component in `Cha` and is separately normalized for a more clear visualization.

space using a CNN as feature extractor. The Siamese network structure means that the weights are shared, which ensures the mathematical requirements for a pseudo-metric. Next, the features are normalized and compared with an element-wise comparison like an absolute or squared difference. Finally, the feature difference is aggregated with sum, mean, and learned weighted average functions. To compute the proposed *VolSiM* metric, that operates on inherently more complex 3D data, the following changes to this framework are proposed.

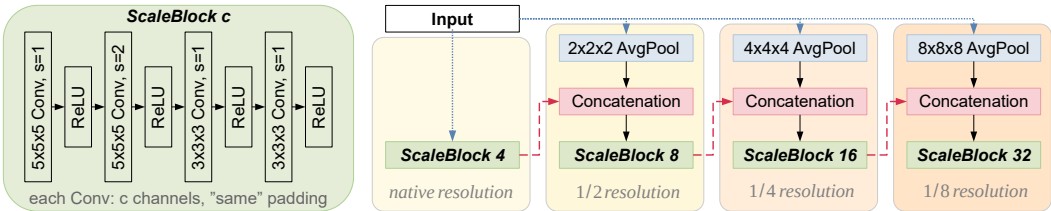

Figure 5: Standard Conv+ReLU blocks (left) are interwoven with input and resolution connections (blue dotted and red dashed), to form the combined network architecture (right) with about $350k$ weights. The output of each scale block is concatenated with the downsampled input for lower scales, leading to features that are spread across multiple resolutions for a stable metric computation.

**Multiscale Network**    Scale is important for a reliable similarity assessment, since physical systems often exhibit self-similar behavior that does not significantly change across scales, as indicated by the large number of dimensionless quantities in physics. Generally, scaling a data pair should not alter its similarity, and networks can learn such an invariance to scale most effectively by processing data at different scales. One example where this is crucial is the energy cascade in turbulence (Pope, 2000), which is also analyzed in our case study below. For learned image metrics, this invariance is also useful (but less crucial), and often introduced with large strides and kernels in the convolutions, e.g. via a feature extractor based on AlexNet (Zhang et al., 2018). In fact, our experiments with similar architectures showed, that models with large strides and kernels generally perform better than models that modify the scale over the course of the network to a lesser extent. However, we propose to directly encode this scale structure in a multiscale architecture for a more accurate similarity assessment, and a network with a smaller resource footprint. Fig. 5 shows the proposed fully convolutional network: Four scale blocks individually process the input on increasingly smaller scales, where each block follows the same layer structure, but deeper blocks effectively cover a significantly larger volume due to the reduced input resolutions. Furthermore, deeper architectures can model complex functions more easily, so we additionally include resolution connections from each scale block to the next resolution level via concatenation. Effectively, the network learns a mixture of connected deep features and similar representations across scales as a result.

**Training and Evaluation**    To increase the model's robustness during training, we used the following data augmentations for each sequence: the data is normalized to $[-1, 1]$, and together randomly flipped and rotated in increments of 90° around a random axis. The velocity channels are randomly swapped to prevent directional biases from some simulations, while scalar data is extended to the three input channels via repetition. For inference, only the normalization operation and the repetition of scalar data is performed. The final metric model was trained with the Adam optimizer with a learning rate of $10^{-4}$ for 30 epochs via early stopping, further details can be found in App. A. To train the metric end-to-end, Alg. 1 is employed to compute the predicted distances $\boldsymbol{d}$ and the ground truth $\boldsymbol{g}$ according to the similarity model. Both are passed to the correlation loss function below, that compares them and provides gradients. To determine the accuracy of a metric during inference, we compute the SRCC between $\boldsymbol{d}$ and $\boldsymbol{w}_s$, where a value closer to 1 indicates a better reconstruction.[4]

**Loss Function**    Given predicted distances $\boldsymbol{d}$ and a ground truth $\boldsymbol{g}$ of size $n$, we train our metric networks with the loss in Eq. 5. It consists of a weighted combination of an MSE and an inverted correlation term $r$, where $\bar{d}$ and $\bar{g}$ denote the mean. While the formulation follows existing work (Kohl et al., 2020), it is important to note that $\boldsymbol{g}$ is computed by our similarity model from Sec. 3, and below we introduce a slicing technique to apply this loss formulation to high-dimensional data sets. For the training process via stochastic gradient descent, Eq. 5 requires a trade-off: A large batch size $b$ is useful to improve training stability via less random gradients for optimization. Similarly, a

---

[4]This is equivalent to $\text{SRCC}(\boldsymbol{d}, \boldsymbol{g})$, since the SRCC measures monotonic relationships and is not affect by monotonic transformations. $\text{SRCC}(\boldsymbol{d}, \boldsymbol{w}_s)$ is just computationally more efficient and has numerical benefits.

sufficiently large value of $n$ is required to keep the correlation values accurate and stable.

$$L(\boldsymbol{d}, \boldsymbol{g}) = \lambda_1(\boldsymbol{d} - \boldsymbol{g})^2 + \lambda_2 \left(1 - \frac{\sum_{i=1}^{n}(d_i - \bar{d})(g_i - \bar{g})}{\sqrt{\sum_{i=1}^{n}(d_i - \bar{d})^2}\sqrt{\sum_{i=1}^{n}(g_i - \bar{g})^2}}\right) \tag{5}$$

However, with finite amounts of memory, choosing large values for $n$ and $b$ is not possible in practice. Especially so for 3D cases, where a single sample can already be memory intensive. In general, $n$ is implicitly determined by the length of the created sequences, via the number of possible pairs within a sequence. Thus, we provide an analysis how the correlation can be approximated in multiple steps for a fixed $n$, to allow for increasing $b$ in return. In the following, the batch dimension is not explicitly shown, but all expressions can be extended with a vectorized first dimension.

The full distance vectors $\boldsymbol{d}$ and $\boldsymbol{g}$ are split in slices with $v$ elements, where $v$ should be a proper divisor of $n$. For any slice $k$, we can compute a partial correlation $r_k$ with

$$r_k = \frac{\sum_{i=k}^{k+v}(d_i - \bar{d})(g_i - \bar{g})}{\sqrt{\sum_{i=k}^{k+v}(d_i - \bar{d})^2}\sqrt{\sum_{i=k}^{k+v}(g_i - \bar{g})^2}}. \tag{6}$$

Note that this is only an approximation, and choosing larger values of $v$ for a given $b$ is always beneficial, if sufficient memory is available. For all slices, the gradients are accumulated during backpropagation since other aggregations would required a computational graph of the original, impractical size. Eq. 6 still requires the computation of the means $\bar{d}$ and $\bar{g}$ as a pre-process over all samples. Both can be approximated with the running

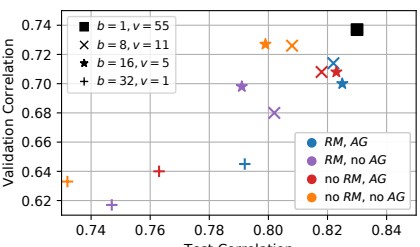

Figure 6: Combined validation and test performance for different batch sizes $b$ and slicing values $v$ (markers), and the usage of a running sample mean $RM$ and correlation aggregation $AG$ (colors).

means $\tilde{d}$ and $\tilde{g}$ for efficiency ($RM$). For small values of $v$, the slicing results in very coarse, unstable correlation results. To alleviate that, it is possible to use a running mean over all previous values $\tilde{r}_k = (1/k)(r_k + \sum_{l=1}^{k-1} r_l)$. This aggregation ($AG$) can stabilize the gradients of individual $r_k$ as they converge to the true correlation value during training, as App. D shows in more detail.

Fig. 6 displays the resulting performance on our data, when training with different combinations of $b$, $v$, $RM$, and $AG$. All models exhibit similar levels of memory consumption and were trained with the same random training seed. When comparing models with and without $RM$ (blue vs red and purple vs orange) both are on par in most cases, even though computation times for a running mean are about 20% lower. The results of models with and without $AG$ (blue vs purple and red vs orange) confirm the expected stabilization effect, where reduced fluctuations in the training translate to better generalizing metrics. Overall, this experiment demonstrates that choosing larger $v$ consistently leads to better results (marker shape), so more accurate correlations are beneficial over a large batch size $b$ in memory limited scenarios. Thus, we use $b = 1$ and $v = 55$ for the final model.

## 6 RESULTS

We compare the proposed *VolSiM* metric to a variety of existing methods in the upper section of Tab. 1. All metrics were evaluated on the volumetric data from Sec. 4, which contain a wide range of test sets that differ strongly from the training data. *VolSiM* consistently reconstructs the ground truth distances from the entropy-based similarity model more reliably than other approaches on most data sets. As expected, this effect is most apparent on the validation sets since their distribution is closest to the training data. But even on the majority of test sets with a very different distribution, *VolSiM* is the best performing or close to the best performing metric. Metrics without deep learning often fall short, indicating that they were initially designed for different use cases, like *SSIM* (Wang et al., 2004) for images, or variation of information *VI* (Meilă, 2007) for clustering. The strictly element-wise metrics *MSE* and *PSNR* exhibit almost identical performance, and both work poorly on a variety of data sets. The learning-based methods *LPIPS* (Zhang et al., 2018) for images, and *LSiM* (Kohl et al., 2020) are only available for two-dimensional data. Hence, they can not correlate structures along all dimensions. To evaluate them in our setting, the 3D data was sliced along all three dimensions to avoid biases against one direction, and the resulting distances are averaged over all slices. As shown in Tab. 1, this results in a reasonable performance overall.

Table 1: Top: performance comparison of different metrics for 3D data via the SRCC, where values closer to 1 indicate a better reconstruction of the ground truth distances. **Bold+underlined** values show the best method for each data set, **bold** values are within a 0.01 margin of the best performing. Bottom: ablation study of the proposed method. Gray models make use of a different training setup.

| | Validation data sets | | | | | Test data sets | | | | | | | | | |
| | Simulated | | | | | Simulated | | Generated | | JHTDB [a] | | | | SF [b] | [c] |
| **Metric** | Adv | Bur | Liq | Smo | | AdvD | LiqN | Sha | Wav | Iso | Cha | Mhd | Tra | SF | All |
|---|---|---|---|---|---|---|---|---|---|---|---|---|---|---|---|
| *MSE* | 0.61 | 0.70 | 0.51 | 0.68 | | 0.77 | 0.76 | 0.75 | 0.65 | 0.76 | **0.86** | **0.80** | 0.79 | 0.79 | 0.70 |
| *PSNR* | 0.61 | 0.68 | 0.52 | 0.68 | | 0.78 | 0.76 | 0.75 | 0.65 | **0.78** | **0.86** | **0.81** | 0.83 | 0.79 | 0.73 |
| *SSIM* | **0.75** | 0.68 | 0.49 | 0.64 | | 0.81 | 0.80 | 0.76 | 0.88 | 0.49 | 0.55 | 0.62 | 0.60 | 0.44 | 0.61 |
| *VI* | 0.57 | 0.69 | 0.43 | 0.60 | | 0.69 | 0.82 | 0.67 | 0.87 | 0.59 | 0.76 | 0.68 | 0.67 | 0.41 | 0.62 |
| *LPIPS (2D)* | 0.63 | 0.62 | 0.35 | 0.56 | | 0.76 | 0.62 | 0.87 | 0.92 | 0.71 | 0.83 | 0.79 | 0.76 | 0.87 | 0.76 |
| *LSiM (2D)* | 0.57 | 0.55 | 0.48 | 0.71 | | 0.79 | 0.75 | 0.93 | **0.97** | 0.69 | **0.86** | 0.79 | 0.81 | **0.98** | 0.81 |
| *VolSiM (ours)* | **0.75** | **0.73** | **0.66** | **0.77** | | **0.84** | **0.88** | **0.95** | 0.96 | **0.77** | **0.86** | **0.81** | **0.88** | 0.95 | **0.85** |
| *CNN_trained* | 0.60 | 0.71 | 0.63 | 0.76 | | 0.81 | 0.77 | 0.92 | 0.93 | 0.75 | 0.86 | 0.78 | 0.85 | 0.95 | 0.82 |
| *MS_rand* | 0.57 | 0.66 | 0.45 | 0.69 | | 0.76 | 0.75 | 0.80 | 0.78 | 0.74 | 0.86 | 0.80 | 0.82 | 0.84 | 0.74 |
| *CNN_rand* | 0.52 | 0.66 | 0.49 | 0.69 | | 0.77 | 0.70 | 0.93 | 0.96 | 0.74 | 0.85 | 0.79 | 0.83 | 0.95 | 0.81 |
| *MS_no skip* | 0.80 | 0.70 | 0.78 | 0.75 | | 0.86 | 0.88 | 0.80 | 0.95 | 0.76 | 0.86 | 0.78 | 0.86 | 0.91 | 0.82 |
| *MS_3 scales* | 0.70 | 0.69 | 0.70 | 0.73 | | 0.83 | 0.82 | 0.95 | 0.94 | 0.76 | 0.87 | 0.80 | 0.88 | 0.93 | 0.83 |
| *MS_5 scales* | 0.78 | 0.72 | 0.78 | 0.78 | | 0.81 | 0.90 | 0.94 | 0.93 | 0.75 | 0.85 | 0.77 | 0.88 | 0.93 | 0.82 |
| *MS_added Iso* | 0.73 | 0.72 | 0.77 | 0.79 | | 0.84 | 0.84 | 0.92 | 0.97 | 0.79 | 0.87 | 0.80 | 0.86 | 0.97 | 0.84 |
| *MS_only Iso* | 0.58 | 0.62 | 0.32 | 0.63 | | 0.78 | 0.65 | 0.72 | 0.92 | 0.82 | 0.77 | 0.86 | 0.79 | 0.65 | 0.75 |

[a] Johns Hopkins Turbulence Database (Perlman et al., 2007)    [b] ScalarFlow data set (Eckert et al., 2019)    [c] Combined test data sets

The bottom half of Tab. 1 contains an ablation study of the proposed architecture *MS*, and a simple *CNN* model. This model is similar to an extension of the convolution layers of AlexNet (Krizhevsky et al., 2017) to three dimension, and does not utilize a multiscale structure. Even though *VolSiM* has more than 80% fewer weights compared to *CNN_trained*, it can fit the training data more easily and works better or equally well on every single dataset in Tab. 1, indicating the strengths of the proposed multiscale architecture. The performance of untrained models *CNN_rand* and *MS_rand* confirm the findings from Zhang et al. (2018), who also report a surprisingly strong performance of random networks. Using no resolution skip connections for *MS_no skip* (see Fig. 5) slightly lowers the metrics generalization across most data sets. Removing the last resolution scale block for *MS_3 scales* overly reduces the capacity of the model, while adding another block for *MS_5 scales* only leads to minor differences. In addition, we also investigate two slightly different training setups: for $MS_{added\ Iso}$ we integrate extra sequences created like the Iso data in the training, while $MS_{only\ Iso}$ is exclusively trained on such sequences. $MS_{added\ Iso}$ only slightly improves upon the baseline, and even the turbulence-specific $MS_{only\ Iso}$ model does not consistently improve the results on the JHTDB data sets. Both cases indicate a high level of generalization for *VolSiM*, as it was not trained on any turbulence data. App. E contains more details for each model and App. F features further ablations.

**Transformation Invariance** Physical systems are often characterized by Galilean invariance (McComb, 1999), i.e. identical laws of motion across inertial frames. Likewise, a metric should be invariant to transformations of the input, meaning a constant distance output when translating, scaling, or rotating both inputs. Element-wise metrics fulfill these properties by construction, but our Siamese network structure requires an equivariant feature representation that changes along with input transformations to achieve them. As CNN features are translation equivariant by design, we only empirically examine rotation and scale invariance for our multiscale metric and a standard Conv+ReLU model on a fixed set of 8 random data pairs from each data set. For the rotation experiment, we rotate the pairs in steps of 5° around a random coordinate axis. The empty volume inside the original frame is filled with a value of 0, and data outside the frame is cut to prevent any scaling.

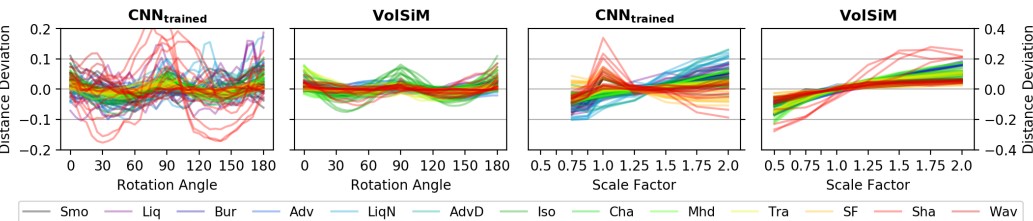

Figure 7: Distance deviation from the mean prediction over different rotation angles (left) and scaling factors of the inputs (right) for a simple CNN and the proposed multiscale model.

For scaling, the data is bilinearly up- or downsampled according to the scale factor, and processed fully convolutionally by each network. In Fig. 7, the resulting distance deviation from the mean of the predicted distances is plotted for rotation and scaling operations. The optimal result would be a perfectly equal distance with zero deviation across all transformations. Compared to the model $CNN_{trained}$, it can be observed that *VolSiM* produces less deviations overall, and leads to significantly smoother distance curves. Note that we observe scale equivariance rather than invariance for *VolSiM*, i.e. a mostly linear scaling of the distances according to the input size. This is not as problematic as the unstable behavior of $CNN_{trained}$, as the general order is preserved and data is typically not compared across resolutions without resampling. The main reason is that the fully convolutional features are larger and thus lead to scale equivariant distances when compared. With a normalization for the physical size of the system, this effect could be further reduced in future work.

**Case Study: Turbulence Analysis**    As a particularly challenging test for generalization, we further perform a detailed case study on forced isotropic turbulence. This study resembles a potential real-world scenario for our metric. For this purpose, fully resolved raw data over a long temporal interval from the isotropic turbulence data set from JHTDB is utilized (see bottom of Fig. 8). The $1024^3$ domain is filtered and reduced to a size of $128^3$ with strides of 8, meaning *VolSiM* is applied in a fully convolutional manner, and has to generalize beyond the training resolution of $64^3$. Three different time spans of the simulation are investigated, where the long span also uses a temporal stride of 20. Traditionally, turbulence research makes use of two-point correlations to study such cases (Pope, 2000). Since we are interested in a comprehensive spatial analysis instead of two single points, we can make use of Pearson's distance on the full fields to obtain reference values for an evaluation. App. G contains further details on the implementation of this case study.

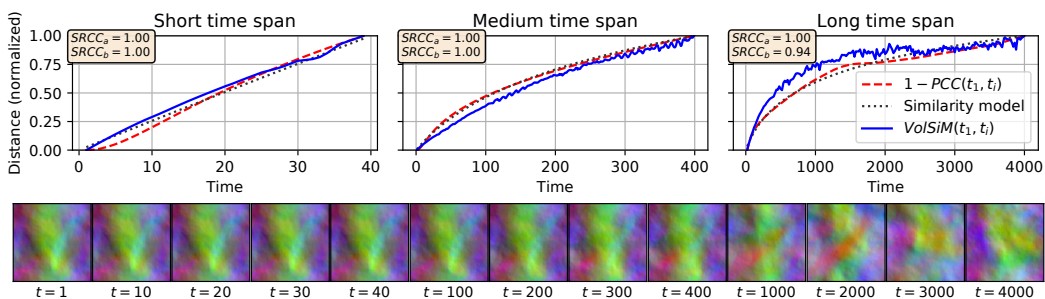

Figure 8: Top: Analysis of forced isotropic turbulence across three time spans. The high SRCC values indicate strong agreement between Pearson's distance and the similarity model (SRCC$_a$), and between the similarity model and *VolSiM* (SRCC$_b$). Bottom: Examples from the overall sequence, visualized via a mean projection along the x-axis and color-coded channels.

At the top of Fig. 8, we find that our similarity model from Sec. 3 (gray dotted) closely follows this correlation-based trajectory (red dashed) over time, as also indicated by the value SRCC$_a$ between both cases. Even though there are smaller fluctuations, the proposed *VolSiM* metric (blue) results in a very similar distance trajectory to the similarity model (see SRCC$_b$) across all time spans. Our similarity metric faithfully recovers the correlation-based reference, despite not having seen any turbulence data at training time. Hence, this experiment indicates the accuracy of the similarity model, and the capabilities for generalization of the multiscale metric network to new cases.

# 7    CONCLUSION

We presented the multiscale CNN architecture *VolSiM*, and demonstrated its capabilities as a similarity metric for volumetric simulation data. A similarity model based on the behavior of entropy in physical systems was proposed and utilized to learn a robust, physical similarity assessment. Different methods to compute correlations inside a loss function were analyzed, and the invariance to scale and rotation transformations investigated. The proposed metric potentially has an impact on a broad range of disciplines where volumetric simulation data arises. An interesting area for future work is designing a metric specifically for turbulence simulations, first steps towards which were taken with our case study. Additionally, investigating learning-based methods with features that are by construction equivariant to rotation and scaling may lead to further improvements in the future.

ETHICS STATEMENT

Since we target the fundamental problem of the similarity assessment of numerical simulations, we do not see any direct negative ethical implications of our work. However, there could be indirect negative effects since this work can act as a tool for more accurate and/or robust numerical simulation methods in the future, for which a military relevance exists. A further indirect issue could be explainability, e.g. when simulations in an engineering process yield unexpected inaccuracies.

REPRODUCIBILITY STATEMENT

To ensure the reproducibility of our work, we took the following steps: An extensive appendix with details and hyperparameters regarding the implementation, data sets, and sequence creation is provided below. Furthermore, the trained *VolSiM* model and source code to evaluate it are provided alongside this submission.

The complete data and implementation of this work will be published upon acceptance. This includes the source code to create the data sets as well as the full data sets themselves, the source code to recreate all figures, and the source code train the metric end-to-end and evaluate it.

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

APPENDIX

In the following, additional details for the proposed *VolSiM* metric are provided: App. A contains implementation details regarding the training and the metric model setup, and App. B features generation details for all our data sets. Next, additional examples of the created data sequences are displayed in App. C, and the training stability for the loss experiment is investigated in App. D. Details for the experimental setup of the ablation study models are given in App. E, and App. F contains additional ablation studies. Finally, details regarding the turbulence case study can be found in App. G.

## A  IMPLEMENTATION DETAILS

The training and evaluation process of the metric was implemented in PyTorch, while the data was simulated and collected with specialized solvers and data interfaces as described in App B. The data acquisition, training, and metric evaluation was performed on a server with an Intel i7-6850 (3.60Ghz) CPU and an NVIDIA GeForce GTX 1080 Ti GPU. It took about 38 hours of training to fully optimize the final *VolSiM* model for the data sequences with a spatial resolution of $64^3$.

In addition to the multiscale feature extractor network, the following operations were used for the Siamese architecture of the metric: Each feature map is normalized via a mean and standard deviation normalization to a standard normal distribution. The mean and standard deviation of each feature map is computed in a pre-processing step for the initialization of the network over all data samples. Both values are fixed for training the metric afterwards. To compare both sets of feature maps in the latent space, a simple element-wise, squared difference is employed. To keep the mathematical metric properties, this also requires a square root operation before the final distance output. The spatial squared feature map differences are then aggregated along all dimensions into a scalar distance output. Here, we used a single learned weight with dropout for every feature map, to combine them to a weighted average per network layer. The activations of the average feature maps are spatially combined with a simple mean, and summed over all network layers afterwards. This process of normalizing, comparing, and aggregating the feature maps computed by the feature extractor follows previous work (Kohl et al., 2020; Zhang et al., 2018).

The weights to adjust the influence of each feature map are initialized to $0.1$, all other weights of the multiscale feature extractor are initialized with the default PyTorch initialization. For the final loss, the MSE term was weighted with $\lambda_1 = 1.0$, while the correlation term was weighted with $\lambda_2 = 0.7$.

## B  DATA SET DETAILS

In the following sections, the details underlying each data set are described. Tab. 3 contains a summary of simulator, simulation setup, varied parameters, noise integration, and used fields for the simulated and generated data sets. Tab. 4 features a summary of the collected data sets, with repository details, jitter and cutout settings, and spatial and temporal $\Delta$ values. Both tables also contain the number of sequences created for training, validation, and testing for every data source. These values only apply for the general metric setup, and changes for the ablation study models can be found in App. E below.

### B.1  ADVECTION-DIFFUSION AND BURGER'S EQUATION

In its simplest form, the transport of matter in a flow can be described by the two phenomena of advection and diffusion. Advection describes the movement of a passive quantity inside a velocity field over time, and diffusion describes the process of dissipation of this quantity due to the second law of thermodynamics.

$$\frac{\partial d}{\partial t} = \nu \nabla^2 d - u \cdot \nabla d \tag{7}$$

Eq. 7 is the simplified Advection-Diffusion equation with constant diffusivity and no sources or sinks, where $u$ denotes the velocity, $d$ is a scalar passive quantity that is transported, and $\nu$ is the diffusion coefficient or kinematic viscosity.

Burger's Equation in Eq. 8 is similar to the Advection-Diffusion equation, but it describes how the velocity field itself changes over time with advection and diffusion. The diffusion term can also be interpreted as a viscosity, that models the resistance of the material to deformations. Furthermore, this variation can develop discontinuities (also called shock waves). Here, $u$ also denotes the velocity and $\nu$ the kinematic viscosity or diffusion coefficient.

$$\frac{\partial u}{\partial t} = \nu \nabla^2 u - u \cdot \nabla u \tag{8}$$

To solve both PDEs, the differentiable fluid framework PhiFlow (Holl et al., 2020) was used. The solver utilizes a Semi-Lagrangian advection scheme, and we chose periodic domain boundary conditions to allow for the usage of a Fourier space diffusion solver. We introduced additional continuous forcing to the simulations by adding a force term $f$ to the velocity after every simulation step. Thus, $f$ depends on the time steps $t$, that is normalized by division of the simulation domain size beforehand. For Adv, Bur, and AdvD, we initialized the fields for velocity, density, and force with multiple layered parameterized sine functions. This leads to a large range of patterns across multiple scales and frequencies when varying the sine parameters.

$$u^x(\boldsymbol{p}) = sum\Big(\boldsymbol{f}_1^x + \sum_{i=1}^{4} \boldsymbol{f}_{i+1}^x * sin(2^i \pi \boldsymbol{p} + c_i \, \boldsymbol{o}_{(i+1\,mod\,2)+1}^x)\Big) \tag{9}$$

$$\text{where } \boldsymbol{c} = (1,\, 1,\, 0.4,\, 0.3)$$

$$f^x(\boldsymbol{p}, t) = sum\Big(\boldsymbol{f}_6^x * (1 + \boldsymbol{f}_6^x * 20) * \sum_{i=1}^{4} \boldsymbol{f}_{i+1}^x * sin(2^i \pi \tilde{\boldsymbol{p}} + c_i \, \boldsymbol{o}_{(i\,mod\,2)+1}^x)\Big) \tag{10}$$

$$\text{where } \tilde{\boldsymbol{p}} = \boldsymbol{p} + \boldsymbol{f}_7^x * 0.5 + \boldsymbol{f}_7^x * sin(3t) \text{ and } \boldsymbol{c} = (0,\, 1,\, 1,\, 0.7)$$

$$d(\boldsymbol{p}) = sum\Big(\sum_{i}^{\{x,y,z\}} sin(\boldsymbol{f}_d^i * 24\pi p_i + o_d^i)\Big) \tag{11}$$

Eq. 9, 10, and 11 show the layered sine functions in $u^x(\boldsymbol{p})$, $f^x(\boldsymbol{p})$, and $d(\boldsymbol{p})$ for a spatial grid position $\boldsymbol{p} \in \mathbb{R}^3$. The $sum$ operation denotes a sum of all vector elements here, and all binary operations on vectors and scalars use broadcasting of the scalar value to match the dimensions. Eq. 12 shows the definition of the function parameters used above, all of which are randomly sampled based on the simulation seed for more diverse simulations. Note that $\nu$ was multiplied by 0.1 for Bur. The remaining velocity and force components $u^y(\boldsymbol{p})$, $u^z(\boldsymbol{p})$, $f^y(\boldsymbol{p})$, and $f^z(\boldsymbol{p})$ and corresponding parameters are omitted for brevity here, since they follow the same initialization pattern as $u^x(\boldsymbol{p})$ and $f^x(\boldsymbol{p})$. Tab. 3 shows the function parameters that were varied, by using the random initializations and adjusting one of them in linear steps to create a sequence.

$$
\begin{aligned}
&\boldsymbol{f}_1^x \sim \mathcal{U}(-0.2, 0.2)^3 && \boldsymbol{f}_2^x \sim \mathcal{U}(-0.2, 0.2)^3 \\
&\boldsymbol{f}_3^x \sim \mathcal{U}(-0.15, 0.15)^3 && \boldsymbol{f}_4^x \sim \mathcal{U}(-0.15, 0.15)^3 \\
&\boldsymbol{f}_5^x \sim \mathcal{U}(-0.1, 0.1)^3 && \boldsymbol{f}_6^x \sim \mathcal{U}(0.0, 0.1)^3 \\
&\boldsymbol{f}_7^x \sim \mathcal{U}(-0.1, 0.1)^3 && \nu \sim \mathcal{U}(0.0002, 0.1002) \\
&\boldsymbol{o}_1^x \sim \mathcal{U}(0, 100)^3 && \boldsymbol{o}_2^x \sim \mathcal{U}(0, 100)^3 \\
&\boldsymbol{f}_d^{x,y,z} \sim \{1, \tfrac{1}{2}, \tfrac{1}{3}, \tfrac{1}{4}, \tfrac{1}{5}, \tfrac{1}{6}\}^3 && \boldsymbol{o}_d^{x,y,z} \sim \mathcal{U}(0, 100)^3
\end{aligned} \tag{12}
$$

The main difference between the Advection-Diffusion training data and the test set is the method of noise integration: For Adv it is integrated into the simulation velocity, while for AdvD it is added to the density field instead. The amount of noise added to the velocity for Bur and Adv and to the density for AdvD was varied in isolation as well.

## B.2 NAVIER-STOKES EQUATIONS

The Navier-Stokes Equations fully describe the behavior of fluids like gases and liquids, via modelling advection, viscosity, and pressure effects, as well as mass conservation. Pressure can be interpreted as the force exerted by surrounding fluid mass at a given point, and the conservation of mass means that the fluid resists compression.

$$\frac{\partial u}{\partial t} + (u \cdot \nabla)u = -\frac{\nabla P}{\rho} + \nu\nabla^2 u + g \qquad (13)$$

$$\nabla \cdot u = 0. \qquad (14)$$

Eq. 13 describes the conservation of momentum, and Eq. 14 describes mass conservation. Again, $u$ denotes the velocity, $P$ is the pressure, $\rho$ is the fluids density, $\nu$ is the kinematic viscosity, and $g$ denotes external forces like gravity.

**Smoke** To create the smoke data set `Smo`, the fluid framework MantaFlow that provides a grid-based Eulerian smoke solver for the Navier-Stokes Equations was used. It is based on a Semi-Lagrangian advection scheme and on the conjugate gradient method as a pressure solver. The simulation setup consists of a cylindrical smoke source at the bottom of the domain with a fixed noise pattern initialization to create more diverse smoke plumes. Furthermore, a constant spherical force field *ff* is positioned over the source. This setup allows for a variation of multiple simulation parameters, like the smoke buoyancy, the source position and different force field settings. They include position, rotation, radius and strength. In addition, the amount of added noise to the velocity can also be varied in isolation.

**Liquid** Both liquid data sets, `Liq` and `LiqN`, were created with a liquid solver in MantaFlow. It utilizes the hybrid Eulerian-Lagrangian fluid implicit particle method (Zhu & Bridson, 2005), that combines the advantages of particle and grid-based liquid simulations for reduced numerical dissipation. The simulation setup consists of two liquid cuboids of different shapes, similar to the common breaking dam setup. After 25 simulation time steps a liquid drop is added near the top of the simulation domain, and it falls down on the water surface that is still moving. Here, the external gravity force as well as the drops position and radius are varied to create similarity sequences. As for the smoke data, a modification of the amount of noise added to the velocity was also employed as a varied parameter. The main difference between the liquid training data and the test set is the method of noise integration: For `Liq` it is integrated into the simulation velocity, while for `LiqN` it is overlayed on the simulation background.

## B.3 GENERATED DATA

To create the shape data set `Sha` and the wave data set `Wav`, a random number of straight paths are created by randomly generating a start and end point inside the domain. It is ensured that both are not too close to the boundaries and that the path has a sufficient length. The intermediary positions for the sequence are a result of linearly interpolating on these paths. The positions on the path determine the center for the generated objects that are added to a occupancy marker grid. For both data sets, overlapping shapes and waves are combined additively, and variations with and without overlayed noise to the marker grid were created.

**Shapes** For `Sha`, random shapes (box or sphere) are added to the positions, where the shape's size is a random fraction of the path length, with a minimum and maximum constraint. The created shapes are then applied to the marker grid either with or without smoothed borders.

**Waves** For `Wav`, randomized volumetric damped cosine waves are added around the positions instead. The marker grid value $m$ at point $\boldsymbol{p}$ for a single wave around a center $\boldsymbol{c}$ is defined as

$$m(\boldsymbol{p}) = cos(w * \tilde{p}) * e^{-(3.7\tilde{p}/r)} \quad \text{where } \tilde{p} = \|\boldsymbol{p} - \boldsymbol{c}\|_2 . \qquad (15)$$

Here, $r$ is the radius given by the randomized size that is computed as for `Sha`, and $w \sim \mathcal{U}(0.1, 0.3)$ is a randomized waviness value, that determines the frequency of the damped wave.

## B.4 COLLECTED DATA

The collected data sets `Iso`, `Cha`, `Mhd`, and `Tra` are based on different subsets from the Johns Hopkins Turbulence Database JHTDB (Perlman et al., 2007), that contain different types of data

from direct numerical simulations (DNS). In these simulations, all spatial scales of turbulence up to the dissipative regime are resolved. The data set SF is based on the ScalarFlow data (Eckert et al., 2019), that contains dense 3D velocity fields of real buoyant smoke plumes, created via multi-view reconstruction technique.

**ScalarFlow** Since 100 reconstructions of different smoke plumes are provided in ScalarFlow, there is no need to add additional randomization to create multiple test sequences. Instead, we directly use each reconstruction sequence to create one similarity sequence in equal temporal steps. The only necessary pre-processing step is cutting off the bottom part of domain that contains the smoke inflow, since it is frequently not fully reconstructed. Afterwards, the data is interpolated to the full spatial size of $128^3$ to match the other data sets.

**JHTDB** The JHTDB subsets typically contain a single simulation with a very high spatial and temporal resolution and a variety of fields. We focus on the velocity fields, since turbulent flow data is especially complex and potentially benefits most from a better similarity assessment. We can mainly rely on using temporal sequences, and only need to add spatial jitters in some cases to increase the difficulty. As turbulence generally features structures of interest across all length scales, we create sequences of different spatial scales for each subset. To achieve this, we randomly pick a cutout scale factor $s$. If $s = 1$, we directly use the native spatial discretization provided by the database. For $s > 1$ we stride the spatial query points of the normal cubical cutout by $s$ after filtering the data. For $s < 1$ the size of the cubical cutout is reduced by a factor of $s$ in each dimension, and the cutout is interpolated to the full spatial size of $128^3$ afterwards. Among other details, Tab. 4 shows the cutout scale factors, as well as the corresponding random weights.

## C  ADDITIONAL EXAMPLE SEQUENCES

Fig. 10, 11, and 12 show multiple full example sequences from all our data sets. In every sequence, the leftmost image is the baseline field. Moving further to the right, the change of one initial parameter increases for simulated data sets, and the spatio-temporal position offset increases for generated and collected data. To plot the sequences, the 3D data is projected along the z-axis to 2D via a simple mean operation. This means, noise that was added to the data or the simulation is typically significantly less obvious due to statistical averaging in the projection. Velocity data is directly mapped to RGB color channels, and scalar data is shown via different shades of gray. Unless note otherwise, the data is jointly normalized to $[0, 1]$ for all channels at the same time, via the overall minimum and maximum of the data field.

## D  TRAINING STABILITY FOR LOSS EXPERIMENT

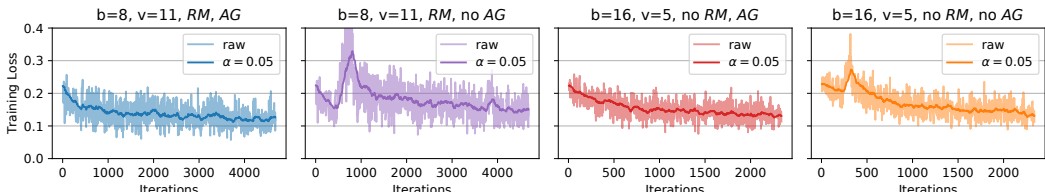

Figure 9: Loss curves from the loss function analysis in Fig. 6 for different models. Shown are the raw losses per batch over all training iterations (lighter line), and an exponential weighted moving average with $\alpha = 0.05$ (darker line) for better visual clarity.

To further analyze the resulting stability of different variants of the loss investigated in Sec. 5, Fig. 9 shows training trajectories of multiple models. Displayed are the direct training loss over the training iterations, i.e. one loss value per batch, and a smoothed version via an exponential weighted moving average computed with $\alpha = 0.05$, corresponding to a window size of 40. The models are color-coded according to Fig. 6, and were trained with different batch sizes $b$, slicing values $v$, usage of the running sample mean $RM$, and usage of the correlation aggregation $AG$. Note that the models were generally trained with the same training seed, meaning the order of samples is should be very similar across all runs (only with potential deviations due to race conditions and GPU processing). It

can be observed that models trained with lower batch sizes generally achieve lower training losses, leading to the better generalization on test and validation sets in Fig. 6. Using no $AG$ makes the training procedure less stable and occasionally large loss spikes occur, leading worse generalization as a result.

## E    ABLATION STUDY DETAILS

In the following, details for the ablation study models in Sec. 6 are provided. The proposed *VolSiM* metric uses the multiscale architecture *MS* described in Sec. 5 as a feature extractor. For all models, the general training setup mentioned in App. A stays identical, apart from: 1) changes to the feature extractor, 2) a different feature extractor, 3) the training amount, or 4) the training data.

**Changes to *MS* Architecture**    For $MS_{no\ skip}$ the resolution skip connections (red dashed arrows in Fig. 5) are entirely removed. This means each resolution is isolated, and sharing information across features of different levels more difficult. For $MS_{3\ scales}$ the last scale block is removed, meaning the architecture from Fig. 5 ends with the $1/4$ resolution level, while the $1/8$ resolution level is omitted. For the $MS_{5\ scales}$ model, one additional scale block is added. In Fig. 5, this corresponds to a $1/16$ resolution level with a $16 \times 16 \times 16$ AvgPool and a scale block with 64 channels.

**Simple CNN Feature Extractor**    The $CNN_{trained}$ model employs an entirely different feature extractor network, similar to the classical AlexNet architecture (Krizhevsky et al., 2017). It consists of 5 layers, each with a 3D convolution followed by a ReLU activation. The kernel sizes (12 - 5 - 3 - 3 - 3) decrease along with the strides (4 - 1 - 1 - 1 - 1), while the number of features first increase, then decreases again (32 - 96 - 192 - 128 - 128). To create some spatial reductions, two $4 \times 4 \times 4$ MaxPools with stride 2 are included before the second and third convolution. The normalization and aggregation of the resulting feature maps is performed as described in App. A, in the same way as for the proposed feature extractor network. As a result, $CNN_{trained}$ can also be applied in a fully convolutional manner for the scaling invariance experiment in Fig. 7.

**Random Models**    For the random models $CNN_{rand}$ and $MS_{rand}$, the corresponding feature extractor along with the aggregation weights are only initialized and not trained. However, to normalize every feature to a standard normal distribution, the training data is processed once to determine the feature mean and standard deviation before evaluating the models, as detailed in App. A.

**Different Training Data**    For the $MS_{added\ Iso}$ model, we created 400 additional sequences from the JHTDB according to the Iso column in Tab. 4, that are added to the other training data. Note that they utilize different random seeds than the 60 test sequences, so the Iso data set becomes an additional validation set in Tab. 1. Similarly for the $MS_{only\ Iso}$ model, 1000 sequences according to the Iso column in Tab. 4 were collected. They replace all other training data from the original model, meaning the Iso data set becomes a validation set, and Adv, Bur, Liq, and Smo become further test sets in Tab. 1. For consistency, the All column in Tab. 1 still reports the combined values of the original test sets for both cases.

## F    ADDITIONAL ABLATIONS

To further investigate our method, we perform three additional ablations in the following. The first ablation focuses on the benefits of our entropy-based similarity model, by replacing it with a simplified identity transformation. The other two replace the multiscale aspect in different ways, to demonstrate the robustness and memory efficiency of the proposed architecture. Tab. 2 shows the resulting SRCC values for these models and the corresponding baselines on our data sets, which are computed as for Tab. 1.

**Simplified Similarity Model**    We compare the *VolSiM* model as proposed in the main paper with $MS_{identity}$ that does not make use of the similarity model based on entropy during training. For that, the logarithmic transformation of $\boldsymbol{w}_s$ is replaced with an identity transformation, corresponding to linear ground truth distances, i.e. $\boldsymbol{g} \leftarrow \boldsymbol{w}_s$ in Alg. 1. The results in the first block in Tab. 2 indicate the benefits of the entropy-based ground truth, with improvements across most data sets. Note that the simplified network version is slightly more efficient as the computation of the parameter $c$ in Alg. 1 can be omitted. However, pre-computing these values for all sequences is computationally very light compared to training the full metric network.

**Singlescale Model**    To investigate the multiscale aspect of our architecture, we removed all scale blocks at lower resolutions, such that only the component at the native input resolution remains (see Fig. 5). To compensate for the lower number of network parameters, we scale the number channels in each layer by a factor of 11, leading to the rather shallow but wide network $MS_{1\ scale}$ with around $360k$ weights. The original network structure $MS_{4\ scales}$ with the four scale blocks reconstructs the ground truth more accurate for almost all data sets as displayed in the middle block of Tab. 2. Furthermore, the proposed structure requires about five times less memory during training.

**No Pooling Layers**    A different approach to analyze the multiscale aspect, is to eliminate all Avg-Pool layers and set all convolution strides to 1 (also see Fig.5). Here, no further adjustments to the network are required as neither pooling layers nor strides alter the number of network weights. The resulting $MS_{no\ pool}$ model is only slightly worse compared to the baseline, as indicated by bottom block in Tab. 2. However, the $MS_{with\ pool}$ model needs about ten times less memory compared to the adjust variant during training due to the significantly smaller feature sizes.

Note that for both ablations on the multiscale structure, the training and test data resolution was reduced to $32^3$. Furthermore, the number of channels in each layer for the $MS_{with\ pool}$ and $MS_{no\ pool}$ models was reduced by a factor of $0.5$. Both choices are motivated by memory limitations and are the reason for the slightly different results across the three baseline models in Tab. 2.

Table 2: Performance comparison of further ablation study models via the SRCC. Shown are a model trained with ground truth distances where the proposed logarithmic similarity model was replaced with an identity transformation (top block), and two models without a multiscale architecture (middle and bottom block). The upper row in each block is the corresponding baseline.

| | Validation data sets | | | | | Test data sets | | | | | | | | | |
|---|---|---|---|---|---|---|---|---|---|---|---|---|---|---|---|
| **Metric** | Adv | Bur | Liq | Smo | | AdvD | LiqN | Sha | Wav | Iso | Cha | Mhd | Tra | SF | All |
| *VolSiM* | 0.75 | 0.73 | 0.66 | 0.77 | | 0.84 | 0.88 | 0.95 | 0.96 | 0.77 | 0.86 | 0.81 | 0.88 | 0.95 | 0.85 |
| $MS_{identity}$ | 0.75 | 0.71 | 0.68 | 0.73 | | 0.83 | 0.85 | 0.87 | 0.96 | 0.74 | 0.87 | 0.77 | 0.87 | 0.94 | 0.82 |
| $MS_{4\ scales}$ | 0.74 | 0.74 | 0.86 | 0.77 | | 0.82 | 0.83 | 0.93 | 0.97 | 0.77 | 0.88 | 0.82 | 0.89 | 0.95 | 0.84 |
| $MS_{1\ scale}$ | 0.73 | 0.71 | 0.78 | 0.75 | | 0.83 | 0.76 | 0.91 | 0.96 | 0.73 | 0.87 | 0.78 | 0.88 | 0.94 | 0.82 |
| $MS_{with\ pool}$ | 0.70 | 0.70 | 0.76 | 0.72 | | 0.82 | 0.79 | 0.94 | 0.96 | 0.76 | 0.83 | 0.79 | 0.88 | 0.97 | 0.83 |
| $MS_{no\ pool}$ | 0.73 | 0.70 | 0.72 | 0.71 | | 0.83 | 0.77 | 0.94 | 0.95 | 0.76 | 0.87 | 0.78 | 0.86 | 0.90 | 0.82 |

# G    TURBULENCE CASE STUDY DETAILS

For the case study in Sec. 6, the velocity field from the *isotropic1024coarse* data set from the JHTDB (Perlman et al., 2007) is utilized. Instead of creating adjusted sequences according to the similarity model, the data is directly converted to three sequences of different time spans without any randomization in this case. A spatial stride of 8 is employed is employed for all cases, and to reduce memory consumption an additional temporal stride of 20 is used for the long time span only. The resulting sequences exhibit a spatial resolution of $128^3$, with 40 frames for the short span, 400 frames for medium span, and 200 frames for the long span. Before further processing, each frame is individually normalized to $[-1, 1]$. To create the results in Fig. 8, the first simulation frame $t_1$ is compared to all following simulation frames $t_i$ via Pearson's distance $1 - PCC(t_1, t_i)$ (red dashed trajectory) and $VolSiM(t_1, t_i)$ (blue solid trajectory). Note that *VolSiM* is applied fully convolutionally here, as it was trained on $64^3$ data. For both cases, the resulting distances are normalized to $[0, 1]$ to visually compare them more easily. The gray dotted curve, is the direct application of the proposed similarity model via Alg. 1 by fitting $c$ for each time span, as described in Sec. 3. The examples from the sequences are visualized as described in App. C, while a projecting along the x-axis here.

Table 3: Data set detail summary for the simulated and generated data sets.

| | Adv | Bur | Liq | Smo | AdvD | LiqN | Sha | Wav |
|---|---|---|---|---|---|---|---|---|
| Sequences train–val–test | 398–57–0 | 408–51–0 | 405–45–0 | 432–48–0 | 0–0–57 | 0–0–30 | 0–0–60 | 0–0–60 |
| Equation | Eq. 7 | Eq. 8 | Eq. 13, 14 | Eq. 13, 14 | Eq. 7 | Eq. 13, 14 | — | — |
| Simulator | PhiFlow [d] | PhiFlow [d] | MantaFl. [e] | MantaFl. [e] | PhiFlow [d] | MantaFl. [e] | MantaFl. [e] | MantaFl. [e] |
| Simulation setup | layered sines | layered sines | breaking dam + drop | rising plume with force field | layered sines | breaking dam + drop | random shapes | random damped waves |
| Time steps | 120 | 120 | 80 | 120 | 120 | 80 | — | — |
| Varied aspects | $f_1, f_2,$ $f_3, f_4,$ $f_5, f_7,$ $o_1, o_2,$ $o_d,$ $noise$ | $f_1, f_2,$ $f_3, f_4,$ $f_5, f_7,$ $o_1, o_2,$ $noise$ | $drop_x$ $drop_y$ $drop_z$ $drop_{rad}$ $grav_x$ $grav_y$ $grav_z$ $noise$ | $buoy_x$ $buoy_y$ $ff_{rot\,x}$ $ff_{rot\,z}$ $ff_{str\,x}$ $ff_{str\,z}$ $ff_{pos\,x}$ $ff_{pos\,y}$ $ff_{rad}$ $source_x$ $source_y$ $noise$ | $f_1, f_2,$ $f_3, f_4,$ $f_5, f_7,$ $o_1, o_2,$ $o_d,$ $noise$ | $drop_x$ $drop_y$ $drop_z$ $drop_{rad}$ $grav_x$ $grav_y$ $grav_z$ $noise$ | shape position | wave position |
| Noise integration | added to velocity | added to velocity | added to velocity | added to velocity | added to density | overlay on non-liquid | overlay on marker | overlay on marker |
| Used fields | density | velocity | velocity flags levelset | density pressure velocity | density | velocity | marker | marker |

[d] PhiFlow from Holl et al. (2020)  [e] MantaFlow: http://mantaflow.com/

Table 4: Data set detail summary for collected data sets.

| | Iso | Cha | Mhd | Tra | SF |
|---|---|---|---|---|---|
| Sequences train–val–test | 0–0–60 | 0–0–60 | 0–0–60 | 0–0–60 | 0–0–100 |
| Repository | JHTDB – isotropic 1024coarse [f] | JHTDB – channel [f] | JHTDB – mhd1024 [f] | JHTDB – transition_bl [f] | ScalarFlow [g] |
| Repository size [h] $s \times t \times x \times y \times z$ | $1 \times 5028 \times$ $1024 \times$ $1024 \times 1024$ | $1 \times 4000 \times$ $2048 \times$ $512 \times 1536$ | $1 \times 1024 \times$ $1024 \times$ $1024 \times 1024$ | $1 \times 4701 \times$ $10240 \times$ $1536 \times 2048$ | $100 \times 150 \times$ $100 \times 178 \times$ $100$ [i] |
| Temporal offset $\Delta_t$ | 180 | 37 | 95 | 25 | 13 |
| Spatial offset $\Delta_{x,y,z}$ | 0 | 0 | 0 | 0 | 0 |
| Spatial jitter | 0 | 0 | 25 | 0 | 0 |
| Cutout scales | 0.25, 0.5, 0.75, 1, 2, 3, 4 | 0.25, 0.5, 0.75, 1, 2, 3, 4 | 0.25, 0.5, 0.75, 1, 2, 3, 4 | 0.25, 0.5, 0.75, 1, 2 | 1 |
| Cutout scale random weights | 0.14, 0.14, 0.14, 0.16, 0.14, 0.14, 0.14 | 0.14, 0.14, 0.14, 0.16, 0.14, 0.14, 0.14 | 0.14, 0.14, 0.14, 0.16, 0.14, 0.14, 0.14 | 0.14, 0.14, 0.14, 0.30, 0.28 | 1 |
| Used fields | velocity | velocity | velocity | velocity | velocity |

[f] JHTDB from Perlman et al. (2007)  [g] ScalarFlow from Eckert et al. (2019)
[h] simulations $s \times$ time steps $t \times$ spatial dimensions $x, y, z$   [i] cut to $100 \times 150 \times 100 \times 160 \times 100$ (removing 18 bottom values from y), since the smoke inflow at the bottom is not fully reconstructed

Adv: Advection-Diffusion (2×density)

Bur: Burger's Equation (2×velocity)

Smo: Smoke (velocity, density, and pressure)

Liq: Liquid (velocity, leveset, and flags)

Figure 10: Example sequences of simulated training data, where each row features a full sequence from a different random seed.

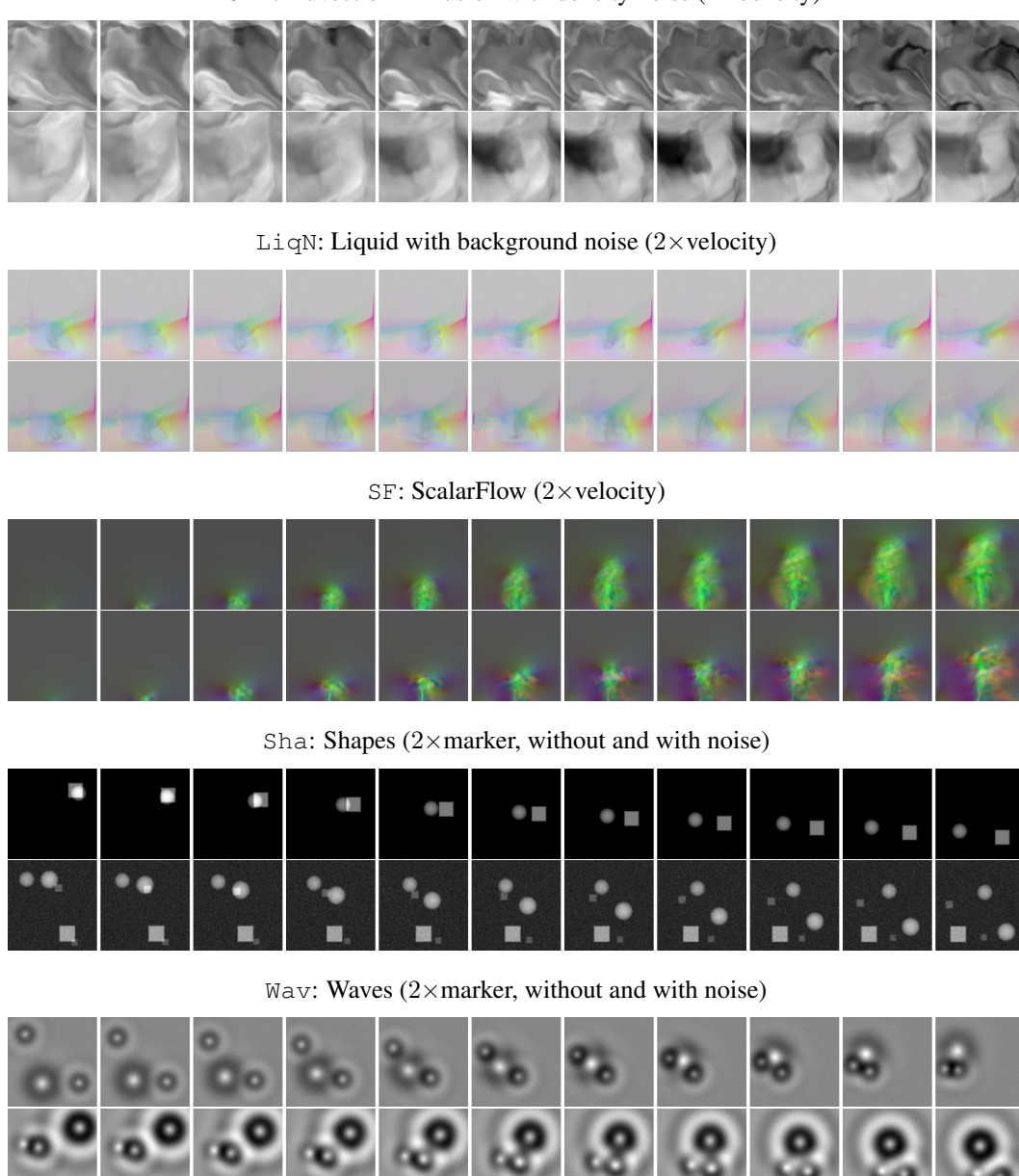

Figure 11: Example sequences of simulated (top two data sets), collected (middle data set), and generated (bottom two data sets) test data. Each row contains a full sequence from a different random seed.

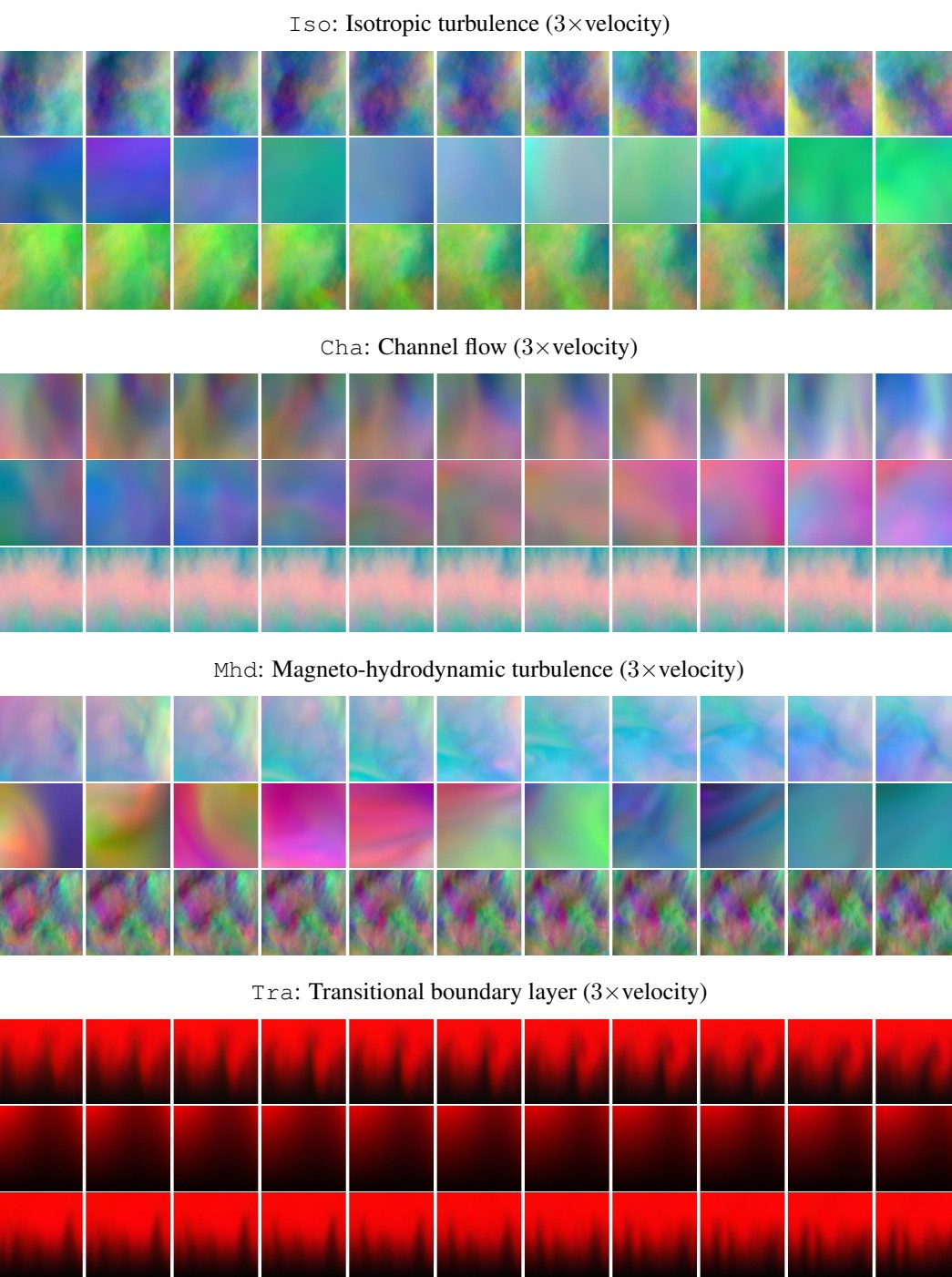

Figure 12: Example sequences of collected test data from JHTDB, where each row shows a full sequence from a different random seed. Notice the smaller cutout scale factor $s$ for the middle example in each case. The predominant x-component in `Cha` and is separately normalized for a more clear visualization.

