# OpenReview forum: "Learning Similarity Metrics for Volumetric Simulations with Multiscale CNNs"
_ICLR.cc/2022/Conference — ICLR 2022 Submitted_

### Official Review · Reviewer_zceM · 2021-10-27

**Correctness:** 4
**Technical Novelty And Significance:** 4
**Empirical Novelty And Significance:** 4
**Recommendation:** 8
**Confidence:** 4

**Main Review:**

The paper is very well written and structured, and it is easy to read. The goal of the paper and its contributions are clearly formulated. The literature review nicely places the proposed work within its field. The argumentations in the paper are well funded and design choices motivated.

The authors included an ethics and reproducibility statement, and announced to publish the complete data and implementation upon acceptance of this paper.

Regarding the equivariance of the proposed CNN. Instead of encouraging the network to learn rotational equivariant features by data augmentation, have you considered to use group convolutions or steerable filters to use a network which is rotational equivariant in every layer?
The plot to explore the rotational equivariance of the metric in Fig. 7 could be interesting to visualize as a polar plot.

Minor comments:
Typo in “[...] against a know ground truth” in the last paragraph of sec. 1.
The abbreviation DNS (Direct Numerical Simulation) is not introduced in the main paper, only in the appendix.





**Summary Of The Paper:**

The paper proposes a similarity model for volumetric simulations derived from Boltzmann’s equation. A formulation of a similarity measure between a reference state and any other microstate in a dissipative physical system is presented. This similarity cannot be used as a metric as is, but can be used to train a Siamese CNN (VolSiM) that is introduced in this paper. The authors formulate a learning task by generating a sequence of physical systems. They use Pearson’s distance to estimate how quickly the microstates in a system are changing, and compute a ground truth for the CNN training using the proposed similarity measure. The network is trained to predict the distances. The CNN is designed to preserve a certain degree of translational, rotational and scale equivariances. Translational equivariance, apart from pooling operations, is given by the nature of CNNs; scale equivariance is directly encoded in the architecture of VolSiM by using four blocks processing the input data in multiple scales; and rotational equivariance is encouraged by data augmentation. The network is given ground truth distances based on the proposed similarity model, predicts distances and uses a loss which is a weighted sum of an MSE term and a correlation term. The authors propose two approaches to generate data, one which is based on varying the initial conditions of PDEs and using PDE solvers for a number of physical simulations, which is used as training data, and one which is based on spatial and temporal variations, which is used to generate test data. Furthermore, a publicly available database is used for testing. The datasets in the experiments cover a wide range of different physical models.

**Summary Of The Review:**

I believe the contribution is relevant to a large audience, is well and clearly written, and method aspects are clearly argued for. The datasets include a lot of variation, and the results are thoroughly discussed. I think the semi-supervised approach to learn a metric for comparing simulation or measurement data based on the proposed similarity model is useful for a wide range of applications and believe the authors also present a nice line of argumentation for design choices which are applicable to other tasks, hence I recommend accepting the submission.

---

> ### Author Response · Authors · 2021-11-17
> **Response to Reviewer zceM**
>
> We are grateful to reviewer zceM for the encouraging and detailed review, as well as the suggestions for further improvements.
>
> **Equivariance / Invariance:**
>
> This is a good point. We did perform preliminary experiments with feature extractor networks based on convolutions that are equivariant to the SE(3) group from Weiler et al. (3D Steerable CNNs: Learning rotationally equivariant features in volumetric data. NeurIPS 2018). However, we did not further investigate this direction due to two main weaknesses of the method in our setup: First, the resulting networks are quite heavy in terms of memory consumption and computation time, which is especially problematic for the large number of pairs we process in parallel already. Second, training the network to a similar level of performance as that achieved by simple data augmentations turned out to be very difficult, as the training process was very unstable for our setup. Nevertheless, it is a very promising direction for further research. We think the theoretical advantages of such methods potentially also translate to direct improvements to our metric if these issues are overcome.

---

> > ### Comment · Reviewer_zceM · 2021-11-18
> > **Author's response**
> >
> > I thank the authors for addressing my questions!

---

### Official Review · Reviewer_nzbf · 2021-11-02

**Correctness:** 3
**Technical Novelty And Significance:** 3
**Empirical Novelty And Significance:** 3
**Recommendation:** 8
**Confidence:** 4

**Main Review:**

Strengths:
The paper is very well written and easy to follow. The proposed entropy-based similarity model is new and appears to be effective in evaluating the similarity between volumetric data in presence of rotation and scaling. The proposed method seems to be the first learning-based similarity measurement methods that compares 3D data generated by a physical system.

Weaknesses:
Overall I'm positive about this paper. But there're still some aspects of the method that I hope the authors could clarify in the rebuttal.

1. Simulated sequence for distance learning. It appears to me that the validity of the proposed similarity measurement relies highly on the simulated sequence. The paper presents an iterative method for determining the simulation step \Delta. However, little is mentioned on how to determine the length n of the simulated sequence. Will different sequence length affect the effectiveness of the similarity model? (which appears to be so based on my understanding.) If so, then how to decide the optimal sequence length for different input data?
Also, in order to simulate the data sequence, the underlying physical mechanism needs to be known. This limits the method to be only applicable to comparing data of a known physical process. In addition, if the simulation model does not match the actual process well, I suspect that the validity of the similarity model will be downgraded.

2. It's not clear how to determine the parameter v (number of elements in vector slices). The current determination method seems be to quite empirical (test on several combinations of b and v). It's not clear whether such combination is universally good for all kinds of volumetric data.

3. Some ablations on the network are missing. In order to prove the effectiveness of the multiscale structure, performance the proposed network should be compared with the baseline network that only uses the native resolution (without the multiscale components). Also some ablations should be done on different sequence lengths and steps.

**Summary Of The Paper:**

The paper presents a learning-based method for evaluating the similarity between 3D volumetric data. A new similarity model based on the entropy of a physical system is proposed, which in my opinion is the paper's major contribution. The method combines the new entropy-based model with a Siamese network that is commonly used for similarity measurement on 2D imagery data. The proposed method further modifies the Siamese network in order to process 3D data. Experimental results are demonstrated on turbulence flow data, and are compared with other 2D similarity measurements.

**Summary Of The Review:**

The paper has good quality. The proposed entropy-based similarity model is novel and seems to be useful. Although some technical details need to be further clarified, overall the work is nice and worth to be presented in ICLR.

---

> ### Author Response · Authors · 2021-11-17
> **Response to Reviewer nzbf**
>
> We thank reviewer nvbf for the positive assessment, and answer the raised questions in the following.
>
> **1a. Sequence length:**
>
> Within certain bounds, the sequence length $n$ does not significantly influence the validity of the trained metric according to our experience. The main reason is that $\Delta$ determines the general coverage of the physical phenomenon within one sequence, while $n$ more or less acts like a discretization of the covered space. Like for other discretizations, choosing a very small $n$ only paints a coarse picture of the underlying phenomenon. However, this could even be alleviated by a significantly higher number of sequences for training. For $n=2$, our problem setting is similar to learning with triplet losses as used for metrics within the area of image classification. In this case, the simulated data is only binarily classified as dissimilar or similar, instead of a more gradual similarity spectrum.
>
> Performing a full ablation study of $n$ to confirm this is unfortunately immensely heavy in terms computational cost: It requires a full resimulation of the data set multiple times, while also recalibrating $\Delta$ in the process to ensure the same range of the physical process is captured for a fair comparison. Nevertheless, this is an interesting area for future investigations.
>
> **1b. Physical mechanism:**
>
> For simulation-based sequences (via method [A]) the underlying physical process must be understood well enough to simulate it. If the simulation does not match the physical process well, the similarity model might indeed only have reduced validity for the true phenomenon. However, comparisons within data from the (potentially inaccurate) simulation are still valid as the network was also trained on these suboptimal cases.
>
> But if there are no (accurate) simulations of a physical process, generation method [B] still allows for the creation of a similarity model, given that the process can be measured with sufficient accuracy. The measurements can be used as the spatio-temporal data repository from which sequences are created. One example that goes towards this direction is the ScalarFlow data which is the basis for our $\texttt{SF}$ test set. For ScalarFlow, real smoke plumes were captured (via videos from different angles) and full 3D flow fields were created via multiple view reconstructions. While this is technically not a pure measurement, the reconstruction process can be considered as an accuracy improvement for the measurement. So in this case, no direct simulations of the underlying phenomenon are involved.
>
> **2. Number of slicing elements:**
>
> In general, while keeping the batch size $b$ constant, picking the largest possible value for $v$ (i.e. the number of pairs within a sequence) is always the preferred option. This integrates all information that is available and yields fully accurate correlation values. The goal of our investigation was not necessarily to empirically find a suitable value of $v$, but to analyze if a lower correlation accuracy in turn pays off for choosing a higher batch size in a memory-limited scenario. We will update the paper to additionally emphasize this.
>
> Here, our choices for $v$ were proper divisors of the number of pairs for a fair comparison that does not discard any data (although other values are also possible). Depending on each $v$, the values of $b$ were chosen to achieve similar memory consumption levels between the networks.
>
> **3. Further ablations:**
>
> We would be happy to provide additional ablations to illustrate the behavior of the multiscale architecture. For this, we created two variants without a multiscale structure:
> First, all scale blocks at lower resolution levels are removed such that only the native resolution component remains (see Fig. 5). To keep the number of weights in the network comparable, the number of channels in the remaining block is multiplied by a factor of 11 leading to *360k* weights. Second, we eliminate the multiscale aspect by removing all AvgPool layers and setting the convolution stride to 1 in all blocks. Due to memory limitations, we multiply the number of channels in each layer by 0.5 in the latter setting, and reduce the data to a resolution of $32^3$ for both cases.
>
> For the first case, we find that the proposed architecture works significantly better across our data sets. The modified architecture without the multiscale aspect results in a 0.02 lower combined correlation value on all test sets compared to the baseline, while using about five times more memory during training. For the second case, the performance difference is smaller but still noticeable. Here, we find a 0.01 lower combined correlation for the modified network, while our architecture also requires about ten times less memory. A detailed description of these models, and a selection of more comprehensive results on this will be included in a revised version of our paper.

---

### Official Review · Reviewer_f7VS · 2021-11-02

**Correctness:** 3
**Technical Novelty And Significance:** 2
**Empirical Novelty And Significance:** 2
**Recommendation:** 3
**Confidence:** 3

**Main Review:**

The core idea of using one-parameter families of states as ground-truth is nice, but this idea appears to be already present in Kohl et al. 2020. I do not follow the motivation for hypothesizing that similarity should behave like entropy. Classical statistical physics posits entropy as a state function for states *at equilibrium,* where generally no macroscopic physical variation is discernable. When comparing, e.g., velocity fields of a fluid that have complicated macroscopic dynamics, why is a statistical physics perspective relevant? What are the "microstates," and how are they related to the macroscopic dynamical variables over which the metric is being defined? Why is it reasonable or necessary to impose a logarithmic prior on similarity between states in one-parameter families? Is there any advantage to this choice over any other monotonic function of the parameter?

The logarithmic hypothesis requires the fitting of a parameter $c$, which is then deduced by using a different similarity metric, PCC, as a proxy. If the authors propose their metric as superior, why does the method rely on PCC as part of its construction of the ground truth?

Perhaps most importantly, the empirical results are underwhelming, with simple metrics like MSE and PSNR often coming out on top of the learned ones. Overall, it is not clear to me what technical or empirical novelty the authors demonstrate over past works such as Kohl et al. 2020.

### Miscellaneous ###
- Algorithm 1 could be expressed more clearly and concisely with formulae rather than an algorithm.
- "our iteration schemes only calibrate $\Delta$ to a suitable magnitude, compared to using a full rejection sampling approach." What do you mean by full rejection sampling? What is being sampled? And if this approach would be better, why not use it?
- It seems from section 5 that the authors use "equivariance" to mean "invariance," but then do not observe scale-invariance. The justification for invariance is that "physical systems exhibit Galilean invariance." But the Galilean group does not include scalings, and many systems are not scale-invariant. It seems reasonable to allow scale-equivariance rather than invariance. Section 5 could be cleaned up to clarify this story.

**Summary Of The Paper:**

This paper describes a learned similarity metric for spatial scalar and vector fields obtained from physics simulations. The metric is like a perceptual loss but for physical fields. The training is self-supervised by comparing one-parameter families of fields obtained by modifying initial conditions or restricting or resampling.

**Summary Of The Review:**

The core idea of using one-parameter families of states as ground-truth is nice, but this idea appears to be already present in Kohl et al. 2020. Overall, it is not clear to me what technical or empirical novelty the authors demonstrate over past works such as Kohl et al. 2020.

---

> ### Author Response · Authors · 2021-11-17
> **Response to Reviewer f7VS [Part 2]**
>
> **Role of the PCC:**
>
> The crucial point here is that the training is not performed against the PCC; the ground truth distances solely come from the entropy-based similarity model itself. The PCC is only employed to calibrate the amount of non-linearity in the similarity model, i.e. as an approximation of how quickly the system changes. Choosing the PCC specifically, is motivated by autocorrelation from turbulence research, which is widely used to indicate how strongly a system is related to a reference state. When using the PCC as a proxy function in the proposed way, it is ensured that it only represents this degree of decorrelation within the system, without affecting the expressiveness of the ground truth distances beyond that.
>
> **Novelty:**
>
> While our work clearly builds upon the findings from Kohl et al. ‘20, we present several novel contributions that set our work apart. Most notably they include:
> - the entropy-based similarity model
> - a turbulence case-study corresponding to the similarity model
> - the multiscale neural network architecture
> -a detailed analysis of the correlation loss function in terms of splitting the correlation computation to increase the batch size in memory intensive setups
> - an investigation into the metrics stability with respect to scale and rotation transformations.
>
> An additional, fundamental difference is that all previous work on metrics (such as Kohl et al. ‘20) focused on 2D data. However, 3D data that exhibits significantly more complex structures for a similarity assessment and poses practical challenges for training stability and memory consumption. To support the future development of new 3D metrics, our source code and all data sets will be published upon acceptance.
>
> **Algorithm 1:**
>
> We respectfully disagree regarding the use of formulae instead of Algorithm 1. While it may be slightly shorter, clarity and readability would suffer in our opinion. It would require the introduction of a rather unintuitive indexing scheme, especially for $d$ and $w_s$.
>
> **Full rejection sampling:**
>
> By “full rejection sampling” approach, we refer to an alternative data generation process. This means generating sequences with random parameters and only keeping sequences within a given difficulty range. This method is not necessarily better, as some outliers can improve the metrics stability, especially as they are in relatively close proximity to the target range (as visible in Fig. 4) due to the calibration of $\Delta$. In addition, sampling sequences with this method is vastly more expensive from a computational perspective and might even introduce a bias in the data through the selection. We will clarify this in an updated version of the paper.
>
> **Equivariance vs invariance:**
>
> We agree that the usage of equivariance (i.e. $f(t\*x)=t\*f(x)$) and invariance (i.e. $f(t\*x)=f(x)$) should be clarified in our exposition. We thank the reviewer for pointing this out. The main reason for the existing ambiguities is that there is a transformation equivariance of the input with respect to the features of the network, while having a transformation invariance with respect to the distance output of the metric. This argumentation will be adjusted accordingly in a revision. The feature equivariance directly leads to a distance invariance for translations and rotations, due to the Siamese network structure. Currently the network tends towards scale equivariance rather than invariance, as the size of the feature maps changes due to the convolutional processing of the inputs. However, a full invariance could be achieved with an additional measurement of the physical real-world size of the system in future revisions of our work. When added to the network, this measurement allows for normalizing the features accordingly, to distinguish between small upsampled cases and physically larger scenarios.

---

> ### Author Response · Authors · 2021-11-17
> **Response to Reviewer f7VS [Part 1]**
>
> We thank reviewer f7VS for the broad range of feedback on our paper. In the following, our comments to the mentioned issues are provided.
>
> **Entropy:**
>
> > “I do not follow the motivation for hypothesizing that similarity should behave like entropy. Classical statistical physics posits entropy as a state function for states at equilibrium, where generally no macroscopic physical variation is discernable.”
>
> This definition of entropy is correct and applies in a setting of statistical physics. However, our motivation is inspired by the broader interpretation of entropy as a general tool from information theory [1]. In this context, entropy can also be interpreted as a measure of diversity, e.g., via Rényi entropy, a generalised form of Shannon entropy [2]. More recently, entropy has also become popular to measure similarities for fuzzy sets [3]. Note, that all these entropy formulations are equivalent [4], and our approach is also motivated by this more general interpretation. We would be happy to extend the discussion in future revisions. Apart from the physical connection, our choice for using Boltzmann’s entropy formulation specifically was motivated by the simplifying assumption of a uniform microstate distribution. Investigating other entropy formulations for our similarity model is an interesting direction for future research, however.
>
> > “When comparing, e.g., velocity fields of a fluid that have complicated macroscopic dynamics, why is a statistical physics perspective relevant? What are the "microstates," and how are they related to the macroscopic dynamical variables over which the metric is being defined?”
>
> In general, microstates are specific microscopic configurations of the physical system. A macrostate that corresponds to a field in our data is a “collection” of several microstates. The microstates for a given macrostate indicate all possible ways in which the system can achieve that macrostate, i.e. how likely the macrostate is. The second law of thermodynamics predicts that systems tend towards macrostates that have a high number of microstates. As a result, the relative change of the number of microstates between fields does tell us something about the macroscopic behavior. For example, when looking at a simulation of a simple diffusion process like the heat equation, a large relative change indicates that the heat is still unevenly distributed (meaning the diffusion is still in an early phase and the system changes quickly). A smaller change means that the heat is almost fully spread out (meaning the diffusion is mostly complete and the system only barely changes anymore).
>
> > “Why is it reasonable or necessary to impose a logarithmic prior on similarity between states in one-parameter families? Is there any advantage to this choice over any other monotonic function of the parameter?”
>
> This “rate of change” of the system in the diffusion example above provides valuable information for similarity assessments: a quickly changing system decorrelates very fast and soon does not bear any resemblance to the original state any more, while a slowly changing system stays correlated and thus similar over a longer time span. Overall, this means states initially become dissimilar very quickly, but changes later on affect the similarity to a lesser extent. The choice of the logarithm as a similarity prior is the result of the entropy-based reasoning above. In an additional ablation study, we replace this logarithmic choice with a simple identity function. We find that this leads to deteriorations on the results across our data sets, with a 0.03 lower combined correlation value on all test sets compared to our method. This ablation with additional details on the setup will be included in a revised version of the paper. Investigating other concave monotonic functions for this purpose is left for future work.
>
> [1] C. Shannon “A mathematical theory of communication”, The Bell System Technical Journal, pp. 379-423, 1948.
>
> [2] A. Rényi “On measures of information and entropy”, Proceedings of the Fourth Berkeley Symposium on Mathematics, Statistics and Probability, pp. 547–561, 1960.
>
> [3] D. Hu, Z. Hong, Y. Wang, “A New Approach to Entropy and Similarity Measure of Vague Soft Sets”, The Scientific World Journal, vol. 2014.
>
> [4] Natal, Jordão, Ivonete Ávila, Victor B. Tsukahara, Marcelo Pinheiro, and Carlos D. Maciel “Entropy: From Thermodynamics to Information Processing”, Entropy 23, no 10: 1340, 2021.

---

### Author Response · Authors · 2021-11-17
**General Response to all Reviews**

We would like to thank all reviewers for their time and effort, as well as their comments and suggestions. More detailed, individual responses for each reviewer can be found in the respective review thread. All minor points not specifically mentioned in our responses (e.g. typos, corrections to formulations, etc.) will be included in a revision of the paper that will be uploaded soon.

---

> ### Author Response · Authors · 2021-11-22
> **Upload of Paper Revision**
>
> We uploaded a revised version of our work, that integrates the changes mentioned in our answers. Areas with major updates are highlighted in green in the paper. In addition to the textual changes, the values for our main results are updated as we added more training sequences to balance the magnitude of the individual training sets. The resulting differences between the models are slightly bigger, while the central findings of our paper remain unchanged.

---

### Decision · Program_Chairs · 2022-01-20

**Decision:**

Reject

**Comment:**

This paper studies a data-driven similarity metric for physical simulation data, based on entropy rate of a physical system. The authors consider a one-parameter family of spatial fields obtained by varying certain parameter, and use those in a self-supervised setup.
Reviewers were split in this submission. While some reviewers highlighted the novelty in the problem setup and the idea of considering one-parameter families, they also expressed concern about the lack of proper justification of the entropy analogy, as well as doubts on the empirical evaluation. Ultimately, and taking all these considerations into account, the AC believes this work would greatly benefit from another review cycle, by addressing the concerns expressed here. Therefore, the AC recommends rejection at this time.